# Utility of Same-Modality, Cross-Domain Transfer Learning for Malignant Bone Tumor Detection on Radiographs: A Multi-Faceted Performance Comparison with a Scratch-Trained Model

**DOI:** 10.3390/cancers17193144

**Published:** 2025-09-27

**Authors:** Joe Hasei, Ryuichi Nakahara, Yujiro Otsuka, Koichi Takeuchi, Yusuke Nakamura, Kunihiro Ikuta, Shuhei Osaki, Hironari Tamiya, Shinji Miwa, Shusa Ohshika, Shunji Nishimura, Naoaki Kahara, Aki Yoshida, Hiroya Kondo, Tomohiro Fujiwara, Toshiyuki Kunisada, Toshifumi Ozaki

**Affiliations:** 1Department of Medical Informatics and Clinical Support Technology Development, Graduate School of Medicine, Dentistry and Pharmaceutical Sciences, Okayama University, Okayama 700-8558, Japan; 2Science of Functional Recovery and Reconstruction, Graduate School of Medicine, Dentistry and Pharmaceutical Sciences, Okayama University, Okayama 700-8558, Japan; 3Department of Radiology, Juntendo University School of Medicine, Tokyo 113-8431, Japan; 4Milliman, Inc., Tokyo 102-0083, Japan; 5Plusman LCC, Tokyo 103-0023, Japan; 6Graduate School of Environmental, Life Natural Science and Technology, Okayama University, Okayama 700-8530, Japan; takeuc-k@okayama-u.ac.jp; 7Department of Orthopedic Surgery, Graduate School of Medicine, Nagoya University, Nagoya 466-8550, Japan; 8Department of Musculoskeletal Oncology and Rehabilitation, National Cancer Center Hospital, Tokyo 104-0045, Japan; sosaki@ncc.go.jp; 9Department of Musculoskeletal Oncology Service, Osaka International Cancer Institute, Osaka 541-8567, Japan; 10Department of Orthopedic Surgery, Graduate School of Medical Sciences, Kanazawa University, Kanazawa 920-8641, Japan; 11Department of Orthopaedic Surgery, Hirosaki University Graduate School of Medicine, Aomori 036-8563, Japan; 12Department of Orthopaedic Surgery, Kindai University Hospital, Osaka 589-8511, Japan; 13Department of Orthopedic Surgery, Mizushima Central Hospital, Kurashiki 712-8064, Japan

**Keywords:** malignant bone tumors, artificial intelligence, transfer learning, YOLO, radiographs, cross-domain learning, diagnostic imaging

## Abstract

Developing effective artificial intelligence (AI) systems for rare cancers, such as malignant bone tumors, is challenging because of data scarcity. This study investigates a data-efficient strategy called “same-modality, cross-domain” transfer learning. We developed an AI model to detect malignant bone tumors using knee X-rays by leveraging a base model trained on a large dataset of chest X-rays (YOLO-TL). We then compared its performance with that of a model trained from scratch using limited bone tumor data (YOLO-SC). While both models achieved similar overall accuracy, the transfer learning model was significantly better in a clinically crucial aspect: it reduced the number of false alarms (false positives) without missing more tumors. This demonstrates that transferring learning across anatomical regions is effective when the imaging modality (e.g., X-rays) is the same. This method offers a new paradigm for the efficient development of reliable and clinically practical AI diagnostic aids for rare diseases.

## 1. Introduction

Primary malignant bone tumors, particularly osteosarcoma, are rare cancers that predominantly affect children, adolescents, and young adults (AYA) [1,2]. Because prognosis is strongly determined by the presence of metastasis at diagnosis, early detection, and therapeutic intervention are critical [3,4,5]. However, the rarity of these tumors means that many physicians have limited clinical experience, making the initial diagnosis in patients presenting with nonspecific symptoms such as growing pain or exercise-induced pain challenging. Findings on plain radiographs, which are the cornerstone of the initial screening, can be subtle and difficult to discern, sometimes leading to diagnostic delays [6]. A longer interval between symptom onset and treatment initiation increases the probability of lung metastasis [7], which is directly linked to a poor prognosis [8].

Currently, medical image diagnosis using artificial intelligence (AI) and deep learning is rapidly advancing, with a high performance reported for the detection, classification, and localization of abnormalities in chest X-ray (CXR) images [9,10,11]. However, in the domain of rare diseases, such as malignant bone tumors, the burden of preparing a sufficient quantity and quality of annotated data is substantial, demanding data-efficient development strategies that can achieve high performance from a limited number of cases. AI—particularly deep learning—has rapidly advanced image interpretation across radiology, enabling robust detection, classification, and localization tasks while raising new issues around validation and deployment in clinical workflows [12].

Transfer learning (TL) is a core approach to this challenge with the potential to improve data efficiency and generalization performance by reusing feature representations learned from large datasets for different tasks [13,14]. However, when the anatomical target or disease spectrum differs, there is a risk of performance degradation due to a domain shift [15,16]. Transfer learning mitigates data scarcity by leveraging pretrained representations; extensive reviews and recent large-scale benchmarks support its effectiveness in medical imaging, with benefits amplified when pretraining and downstream tasks share modality characteristics [17,18,19,20].

From a data-centric perspective, we previously reported that high-precision annotations by specialists contribute to performance improvement [21] and that a three-class annotation design based on anatomical features contributes to improved specificity [22]. Building on these findings, this study validates the utility of “same-modality, cross-domain” transfer learning, which crosses anatomical regions within the same imaging modality. Specifically, we used a pretrained CXR reading model as the base and performed transfer learning on our dataset of X-ray images of osteosarcoma and Ewing sarcoma of the knee joint. Although the anatomical structures and shadow patterns of the chest and knee are vastly different, we hypothesize that grayscale images based on the same X-ray imaging physics share commonalities in low- to mid-level features such as contrast, edges, density gradients, and scatter noise characteristics. We conducted this study based on the assumption that these commonalities would function as robust starting points to efficiently learn high-level features specific to bone lesions. Within X-ray modalities, emerging evidence underscores the value of modality-aligned pretraining, motivating our same-modality, cross-domain TL design for rare malignant bone tumors [23].

This study utilizes a you only look once (YOLO)-based object detector [24] to construct and compare a model with transfer learning (YOLO-TL) and a scratch-trained model without pretrained weights (YOLO-SC) under identical conditions. We will perform receiver operating characteristic (ROC) and precision–recall (PR) analyses by sweeping the output score threshold and comprehensively evaluate clinical decision-making metrics, such as positive and negative predictive values, likelihood ratios, number needed to screen, and decision curve analysis, at multiple operating points (e.g., high-sensitivity or high-specificity operations), considering the trade-off between sensitivity and specificity. The contributions of this study are threefold: first, to systematically validate the utility of transfer learning across anatomical regions within the X-ray modality for the rare disease of primary malignant bone tumors; second, to directly compare YOLO-TL and YOLO-SC under identical architectures and training conditions to quantify differences in model characteristics that are not captured by a single metric, such as area under the curve (AUC); and third, to visualize the behavior of both models in multiple clinically assumed scenarios to provide insights applicable in practice.

## 2. Materials and Methods

### 2.1. Study Population and Dataset

In this study, two types of datasets were prepared: a training/internal validation dataset for model development and a completely independent external test dataset for performance evaluation.

For positive cases, 468 plain radiographs were obtained from 31 patients with osteosarcoma around the knee joint treated at Okayama University Hospital. As a negative control group, 378 X-ray images of 54 healthy knee joints without obvious degenerative findings, taken at the same hospital, were used. Given that osteosarcoma is a rare disease, to maximize the diversity of images obtained from a limited number of cases, all positive and negative images were pooled together. From this pool, 70% (328 positive, 265 negative) were randomly assigned for model training, and the remaining 30% (140 positive, 113 negative) were used for internal validation. While this method carries a risk of overfitting, as images from the same patient could be included in both training and validation sets, it was adopted as a strategic decision to ensure data diversity. To assess this potential bias, the final performance of the model was evaluated on the completely independent external test dataset described below.

To rigorously evaluate the generalization performance of the models, an independent dataset not used for training or internal validation was employed. For positive cases, 268 radiographs from 32 osteosarcoma cases were provided by the Osaka International Cancer Center. For negative cases, 475 radiographs from 148 healthy knee joint cases were provided by Mizushima Central Hospital.

### 2.2. Model Development and Annotation

In this study, we adopted the YOLO object-detection algorithm as the base architecture. For performance comparison, we constructed two models: YOLO-TL, a model fine-tuned using transfer learning, and YOLO-SC, a model trained from scratch using only a bone tumor dataset. We adopted YOLOv5 (Ultralytics, open-source implementation, https://github.com/ultralytics/yolov5 (accessed on 24 September 2025)) as the base object-detection architecture for both YOLO-TL and YOLO-SC. For both YOLO-TL and YOLO-SC, training was performed in 10 cycles. At the beginning of each cycle, the optimizer was re-initialized while the model weights obtained in the previous cycle were retained, effectively implementing a warm-restart style fine-tuning strategy aiming to stabilize convergence and improve exploration of the loss landscape, similar in spirit to restart-based schedules (e.g., SGDR [25]; and Cyclical Learning Rates [26]). An exponential moving average (EMA) of the model parameters was also maintained throughout training to smooth fluctuations and improve inference robustness. The only methodological difference between YOLO-TL and YOLO-SC was initialization: YOLO-TL started from a chest X-ray–pretrained YOLOv5 model, whereas YOLO-SC was trained from scratch. The annotation for the model’s training data followed the method whose effectiveness was demonstrated in our previous research [21]. Specifically, a musculoskeletal tumor specialist with over 10 years of experience performed precise annotations on the X-ray images by accurately identifying the tumor extent while referencing MRI images.

### 2.3. Statistical Analysis

The following statistical analyses were performed to compare the diagnostic performance of both models in a multi-faceted and quantitative manner. The reporting of this study adheres as much as possible to the Standards for Reporting of Diagnostic Accuracy Studies guidelines for diagnostic accuracy studies and the Checklist for AI in Medical Imaging guidelines for AI research. The statistical significance level was set at a two-sided *p*-value < 0.05, and multiplicity was interpreted descriptively without adjustment. Analyses were conducted using Python 3.11, PyTorch 2.0.1, scikit-learn (version 1.3), and SciPy (version 1.11).

#### 2.3.1. Diagnostic Performance Metrics

For performance evaluation, each image in the external test dataset was labeled as malignant (label “1”) or normal (label “0”). The overall discriminative ability of the models was primarily evaluated using the area under the ROC curve (AUC). To assess performance at specific decision thresholds, we also calculated sensitivity, specificity, positive predictive value (PPV), negative predictive value (NPV), and the F1 score.

#### 2.3.2. Definition of Clinical Operating Points

To compare model performance in clinically relevant scenarios, we predefined three distinct operating points based on the models’ continuous output scores (0 to 1):Youden-optimal point: The threshold that maximizes Youden’s index (Sensitivity + Specificity − 1), representing the optimal balance between sensitivity and specificity.High-sensitivity point: The threshold that maximizes specificity while maintaining a sensitivity of at least 0.90, designed for screening scenarios where minimizing missed cases is the priority.High-specificity point: The threshold that maximizes sensitivity while maintaining a specificity of at least 0.90, designed for scenarios where minimizing false-positive results and subsequent unnecessary workups is crucial.

#### 2.3.3. Statistical Comparison

We used case-level stratified bootstrapping with 5000 resamples to generate 95% confidence intervals (CIs) for all performance metrics using the percentile method. The statistical significance of the difference in AUC between the two models was determined using DeLong’s test. At each predefined operating point, McNemar’s test with continuity correction was used to compare the sensitivity and specificity of the paired model predictions.

#### 2.3.4. Supplementary Performance and Utility Analyses

To provide a more comprehensive evaluation, the following supplementary analyses were performed.

PR Analysis: We generated PR curves and calculated the Average Precision (AP) to evaluate model performance, taking into account the dataset’s class imbalance (prevalence).Calibration Assessment: We assessed the reliability of the models’ predictive scores using calibration plots, generated by plotting observed event rates against predicted probabilities within 10 decile bins. The Brier score was calculated to quantify the overall prediction error.Decision Curve Analysis (DCA): We evaluated the clinical utility of the models by calculating the net benefit across a range of threshold probabilities. DCA visualizes the clinical value of a model by weighing the benefits of true positives against the harms of false positives, allowing for a comparison against default strategies of treating all or no patients.

### 2.4. Ethical Considerations

This study was conducted in compliance with the ethical principles of the Declaration of Helsinki and was approved by the Ethics Committee of Okayama University Graduate School of Medicine, Dentistry, and Pharmaceutical Sciences (Approval No. 2304-004). Informed consent from the subjects was obtained via an opt-out method, wherein information about the research was publicly posted, guaranteeing participants the opportunity to refuse participation.

## 3. Results

This study compared the performance of transfer-learned YOLO-TL and scratch-trained YOLO-SC. The analysis was conducted using an independent external test dataset of 743 X-ray images (268 positive, 36.1%; 475 negative, 63.9%), with all cases included in the primary analysis.

### 3.1. Overall Discriminative Performance of Models: ROC Curve and AUC

First, the overall discriminative performance of both models was evaluated using the ROC curves and the AUC (Figure 1). The AUC for YOLO-SC was 0.961 (95% CI: 0.948–0.973) and that for YOLO-TL was 0.954 (95% CI: 0.937–0.970). There was no statistically significant difference in the AUCs of the two models (Difference [YOLO-SC − YOLO-TL]: +0.0068, 95% CI: −0.0135–+0.0282, DeLong’s test *p* = 0.53). The ROC curves and CI bands for both the models largely overlapped, indicating that their overall discriminative abilities were equivalent (Figure 1).

### 3.2. Diagnostic Performance for Clinical Operation

#### 3.2.1. Sensitivity and Specificity

Next, the sensitivity and specificity of both models were compared using McNemar’s test at three different cutoff values assumed for clinical application (Figure 2). The comparisons of multiple metrics in this section are positioned as exploratory evaluations of prespecified key operating points, and no statistical adjustments for multiplicity were made.

#### 3.2.2. Youden’s Index Maximization Point (Balanced)

YOLO-TL had a sensitivity of 0.914 (TP 245/FN 23) and specificity of 0.897 (TN 426/FP 49), whereas YOLO-SC had a sensitivity of 0.892 (239/29) and specificity of 0.884 (420/55). In the paired binary comparison (McNemar’s test with continuity correction), the differences were not significant (sensitivity difference, *p* = 0.391; specificity difference, *p* = 0.511).

#### 3.2.3. High-Sensitivity Operating Point (Missed-Case-Suppression Focused)

Since preventing missed cases of malignant tumors is the most critical performance requirement for this product, we fixed the sensitivity at ≥0.90 (achieved value 0.903) for comparison. Under these conditions, the specificity of YOLO-TL was 0.903, which was significantly higher than that of YOLO-SC at 0.867 (difference +0.036, *p* = 0.037). This implies that, while maintaining the same number of missed cases, YOLO-TL can reduce false positives by approximately 17 cases per 475 negative cases. The corresponding confusion matrices at this operating point are shown in Appendix A.

#### 3.2.4. High-Specificity Operating Point (False-Positive-Suppression Focused)

To avoid unnecessary follow-up examinations, we fixed the specificity at ≥0.90 (achieved value approx. 0.902) for comparison. Under these conditions, the YOLO-TL sensitivity was 0.798, which was statistically significantly higher than that of YOLO-SC (0.764, difference +0.034, *p* = 0.0077). These results clarify that, although there is no difference in the overall AUC, YOLO-TL exhibits more favorable performance characteristics when the operating point is set according to specific clinical needs.

### 3.3. Detection Performance and Balance for Positive Cases

PR curves and F1 scores were analyzed to evaluate the ability to correctly identify positive cases (Figure 3a,b). The area under the PR curve (AP) was substantially equivalent for both models at 0.944. However, focusing on the operating point with a sensitivity (recall) of 0.903, the precision (PPV) of YOLO-TL was 0.840, which was significantly higher than that of YOLO-SC’s 0.793 (*p* = 0.030). Furthermore, when the F1 score, the harmonic mean of Precision and Recall, was maximized, YOLO-TL showed a higher trend (0.873 vs. 0.853), although this did not reach statistical significance (*p* = 0.074).

At both the high-sensitivity and high-specificity operating points, YOLO-TL was better able to suppress false positives than YOLO-SC (Figure 4).

### 3.4. Reliability of Prediction Scores: Calibration Analysis

A calibration analysis was performed to evaluate the reliability of the score outputs of the models (Figure 5). Comparing the closeness to ideal calibration (slope = 1), the slope for YOLO-TL was 0.865 and that for YOLO-SC was 1.449, with YOLO-TL being significantly closer to 1 (*p* = 0.014). This suggests that the YOLO-TL scores can be interpreted as probabilities with less correction. In contrast, the Brier score, which indicates the average prediction error, was slightly better (lower) for YOLO-SC (0.089 vs. 0.100).

### 3.5. Evaluation of Clinical Utility: DCA

Finally, DCA was conducted to evaluate the net clinical benefit of implementing the models (Figure 6). Both models showed a higher net benefit than the strategies of “treat all” or “treat none” over a wide range of threshold probabilities, confirming their clinical utility as diagnostic support tools. In a direct comparison, YOLO-SC showed a marginally, yet statistically significantly, higher net benefit than YOLO-TL in the threshold probability range of 0.10 to 0.25 (*p* < 0.01), though the difference was minimal.

Taken together, these results indicate that while the transfer-learned YOLO-TL was equivalent to the scratch-trained YOLO-SC in terms of overall discriminative ability (AUC), it showed statistically significant superior performance characteristics in clinically important high-sensitivity or high-specificity operating regions (Table 1).

## 4. Discussion

This study conducted a multi-faceted investigation of the impact of transfer learning on the performance characteristics of an AI model for diagnosing malignant bone tumors on radiographs. The main finding was that although the YOLO-TL model, pretrained on CXR images, showed no statistically significant difference in the overall discriminative ability (AUC) compared to the YOLO-SC model trained from scratch, it demonstrated better performance at specific, clinically important operating points. This discussion interprets the significance of these results based on prespecified evaluation points without adjusting for multiplicity.

### 4.1. The Performance Difference Induced by Transfer Learning and Its Mechanism

The most intriguing finding of this study was that a significant difference in performance emerged at specific operating points despite the lack of difference in the average AUC metric. This can be explained by the effect of transfer learning on the model. Initially, there was concern about “negative transfer” due to the domain shift from chest radiographs, with their vastly different anatomical structures, to bone tumor images. However, no clear negative transfer, such as a significant decrease in overall discriminative ability (AUC), was observed. This suggests that even between different anatomical sites, such as the chest and knee, the common modality of “X-ray imaging” provides sufficient commonality in low- to mid-level features, such as contrast, edges, and density gradients, and the learning effect of this common foundation outweighs the potential negative effects of the domain shift. The general-purpose feature extraction capability acquired from CXRs effectively functioned as a robust initial value for learning lesion-specific features from a limited bone tumor dataset.

It is believed that this benefit from pretraining enables YOLO-TL to achieve a more stable generalization performance. Indeed, in the calibration analysis, the slope of YOLO-TL’s prediction scores (slope = 0.87) was significantly closer to the ideal value of 1 than that of YOLO-SC (slope = 1.45), suggesting a better ability to properly adjust its “confidence” in predictions.

However, YOLO-SC showed a slightly better performance than YOLO-TL in terms of the Brier score and in certain threshold regions of the DCA. This suggests that, because YOLO-SC was trained specifically on a limited bone tumor dataset, its prediction error and net benefit under a specific cost balance may have been “locally optimized.” However, as a trade-off for this specialization, it was inferior to YOLO-TL, which was pretrained on diverse images, in terms of generalization ability to unseen data and reliability of prediction scores (calibration distortion). DCA indicated a small net-benefit advantage for YOLO-SC within a limited probability-threshold range (approximately pt = 0.10–0.25; Figure 6). This may matter in workflows that explicitly apply a threshold-based triage rule. However, our prespecified clinical use case prioritizes a high-sensitivity operating point to avoid missed malignant tumors; in that context, YOLO-TL reduced false positives at identical sensitivity, so we view the DCA difference as modest and context-dependent rather than practice-changing.

In conclusion, this stability and superior generalization performance acquired by YOLO-TL manifests as the ability to more skillfully adjust the trade-off between false positives and false negatives, leading to superior performance at clinically desirable operating points.

### 4.2. Clinical Significance and Operational Implications

The findings of this study have important implications for the clinical implementation of AI. In the screening of rare cancers, such as primary malignant bone tumors, reducing missed cases (high sensitivity) is paramount; however, an increase in false positives leads to physical and psychological burdens on patients and a waste of medical resources; therefore, they must be suppressed as much as possible [27].

This characteristic demonstrated by YOLO-TL significantly reduces false positives while maintaining high sensitivity, which means that it can directly contribute to the efficiency of the diagnostic process by reducing the number of unnecessary follow-up examinations [28,29]. Even in regions where specialists are scarce, it is expected to function as a reliable screening tool, contributing to improved access for early detection and treatment [30,31]. Given that the DCA showed that both models are “worth using,” YOLO-TL, with its superior trade-off performance, can be considered a more readily acceptable model in real-world clinical practice.

### 4.3. Relation to Prior Work and Originality of This Study

Our previous research showed that high-quality annotations by specialists and a three-class annotation based on anatomical features improve AI model performance, particularly specificity [21,22]. The results of this study reaffirm the importance of these “Data-Centric Approaches” [32,33,34,35,36]. That is, providing the model with high-quality training data or data with more information content (in this study, the features from pretraining) leads to more sophisticated diagnostic capabilities, which is consistent with previous findings. In addition, the originality of this study lies in (1) demonstrating the effectiveness of transfer learning between completely different anatomical sites (chest and knee), and (2) revealing through multi-faceted metrics that a difference in clinical utility can exist even between models with no difference in AUC. This presents the possibility of an efficient and practical development strategy that utilizes easily accessible data from other domains to improve model performance in areas where data collection is limited, such as rare diseases.

### 4.4. Limitations and Future Perspectives

This study had several limitations. First, the test dataset was from a limited number of institutions, and large-scale external validation through further multicenter collaborative studies is essential to evaluate the generalization performance of the model more robustly. Second, multiple images from the same patient were treated as independent data points, indicating that clustering by patient units was not strictly considered. Third, the source data for transfer learning in this study were limited to chest X-rays, and the investigation of the optimal source data remains a future task. Future work should include prospective multicenter studies to overcome these limitations, explore methods to explicitly train the model on challenging negative examples such as epiphyseal lines and osteophytes, and investigate new pretraining strategies such as self-supervised learning. Ultimately, it is crucial to establish a diagnostic AI development pipeline that minimizes missed cases while suppressing over-detection through a tripartite strategy of “same-modality transfer learning,” “high-quality annotation,” and “institution-specific calibration.”

## 5. Conclusions

This study demonstrated that same-modality, cross-domain transfer learning, while not improving the overall discriminative performance as measured by the area under the curve (AUC), significantly enhances the clinical utility of an AI model for detecting malignant bone tumors by optimizing performance at clinically critical operating points. The transfer-learned model achieved superior specificity and positive predictive value at a high-sensitivity threshold. This crucial advantage allows for the suppression of false positives while maintaining a high detection rate for malignant lesions. Consequently, this approach offers a direct pathway to reducing unnecessary patient workups and optimizing clinical workflows, thereby enhancing the practical value of the AI as a diagnostic aid.

In conclusion, the value of transfer learning in this context lies not in elevating summary metrics, but in refining model behavior to align with specific clinical demands. This underscores the importance of a comprehensive development strategy centered on high-quality data, appropriate transfer learning, and nuanced operating threshold design to build safe, effective, and clinically trusted AI systems for rare disease diagnostics.

## Figures and Tables

**Figure 1 cancers-17-03144-f001:**
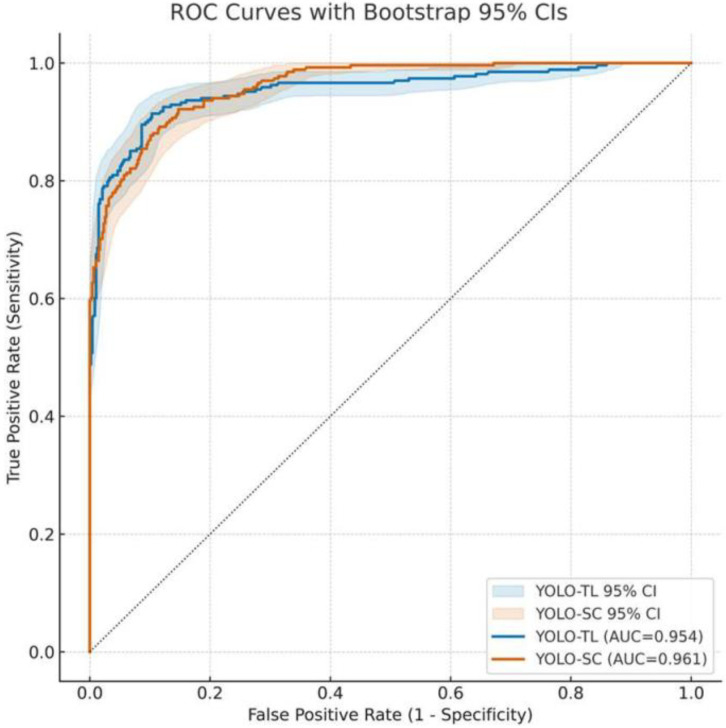
Receiver operating characteristic (roc) curves for the transfer learning and scratch-trained models. The figure presents a comparative analysis of the diagnostic performance between the YOLO-TL model (transfer learning; blue line) and the YOLO-SC model (trained from scratch; orange line) evaluated on the external test dataset. The Area Under the Curve (AUC) was 0.954 (95% Confidence Interval [CI], 0.937–0.970) for YOLO-TL and 0.961 (95% CI, 0.948–0.973) for YOLO-SC. A statistical comparison using DeLong’s test confirmed that the difference in the AUC between the two models was not significant (*p* = 0.53). The shaded ribbons depict the 95% CIs for the ROC curves, which were estimated using a patient-level bootstrap procedure with 5000 re-samples. The dashed diagonal line represents the performance of the random classifier (AUC = 0.5).

**Figure 2 cancers-17-03144-f002:**
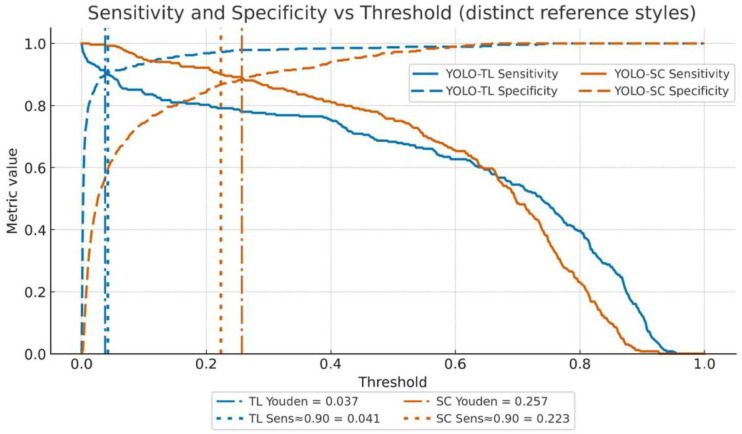
Sensitivity and specificity as a function of the decision threshold. This figure illustrates the performance characteristics of YOLO-TL and YOLO-SC by plotting their sensitivity (solid lines) and specificity (dashed lines) against a continuous range of decision thresholds. Two prespecified clinically relevant operating points are highlighted with vertical lines for each model. The dash-dotted lines indicate the Youden-optimal thresholds, which balance sensitivity and specificity (YOLO-TL: 0.037; YOLO-SC: 0.257). The dotted lines represent the high-sensitivity operating points, where a sensitivity of 0.903 was maintained while maximizing specificity (YOLO-TL, 0.041; YOLO-SC, 0.223). The plot demonstrates that at this high sensitivity threshold, the YOLO-TL model achieved a significantly higher specificity than the YOLO-SC model (0.903 vs. 0.867; *p* = 0.037).

**Figure 3 cancers-17-03144-f003:**
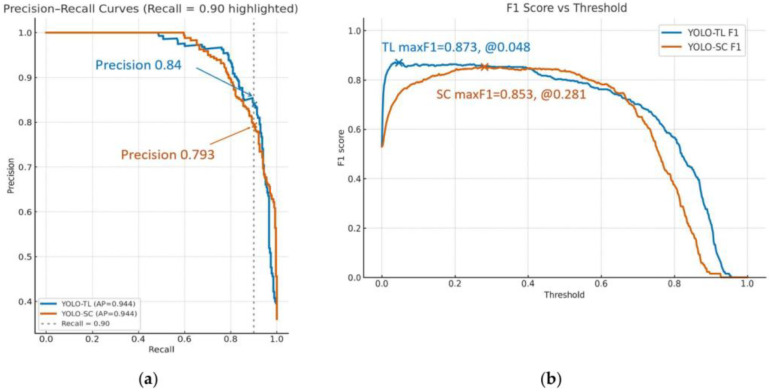
Analysis of positive predictive performance and model balance. (**a**) The precision–recall (PR) curves for the YOLO-TL (blue line) and YOLO-SC (orange line) models are presented. While the overall performance, measured by Average Precision (AP), was nearly identical for both models (AP ≈ 0.944), their characteristics differ at specific operating points. At a high-recall threshold of 0.903 (vertical dotted line), the YOLO-TL model demonstrated a significantly higher precision (Positive Predictive Value) of 0.840 compared to 0.793 for the YOLO-SC model (*p* = 0.030). (**b**) The F1 score, representing the harmonic mean of precision and recall, is plotted as a function of the decision threshold. The analysis identifies the thresholds that yield the maximum F1 score for each model, representing the optimal balance between precision and recall. The YOLO-TL model achieved a maximum F1 score of 0.873 at a threshold of 0.048, whereas the YOLO-SC model reached a maximum of 0.853 at a threshold of 0.281. The difference between these peak F1 scores was not statistically significant (*p* = 0.074).

**Figure 4 cancers-17-03144-f004:**
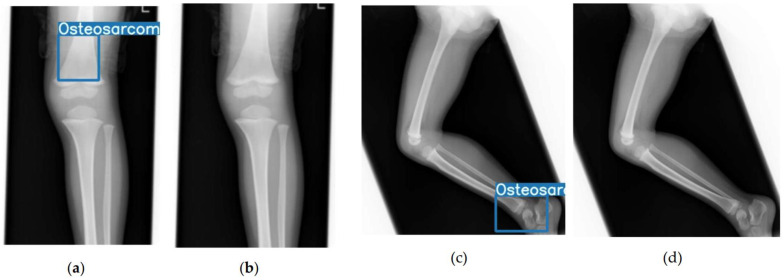
Representative examples of false positive reduction by the transfer learning model. This figure presents two representative cases from the negative (normal) cohort to highlight the key performance differences between the models. Panels (**a**,**c**) show the outputs from the scratch-trained model (YOLO-SC), which incorrectly produced false-positive detections, as indicated by the blue bounding boxes. In contrast, panels (**b**,**d**) show the predictions for the same images from the transfer learning model (YOLO-TL), which correctly classifies them as negative (true negatives). These examples visually demonstrate the improved ability of the YOLO-TL model to avoid erroneous detections in challenging negative cases compared with the YOLO-SC model.

**Figure 5 cancers-17-03144-f005:**
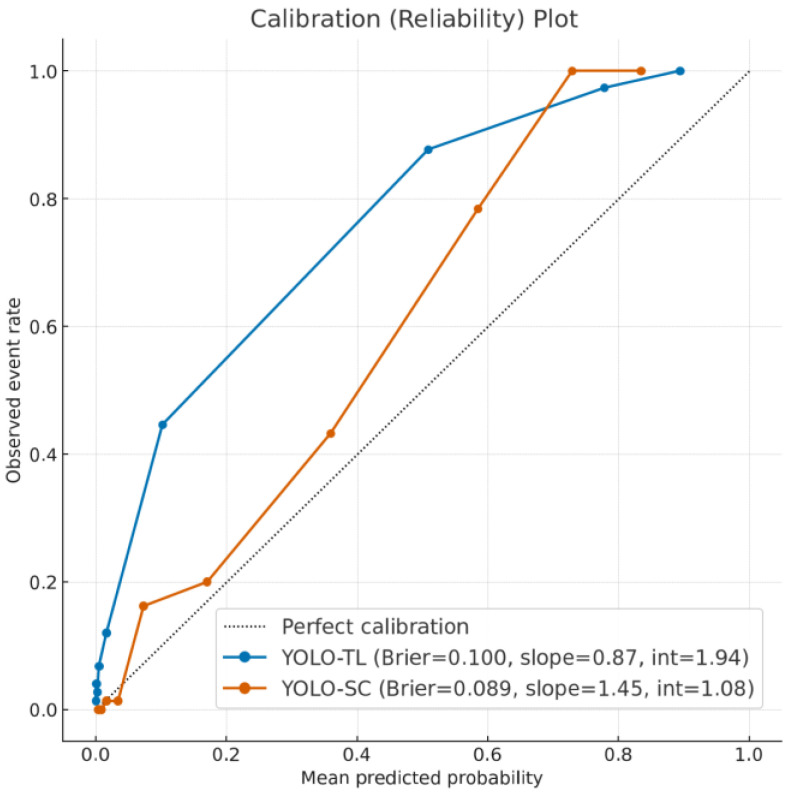
Calibration plots evaluating model reliability. This figure shows the calibration of the YOLO-TL (blue line) and YOLO-SC (orange line) models. For this analysis, the predictions were partitioned into deciles based on the predicted probability (*x*-axis), and the mean predicted probability for each bin was plotted against the corresponding observed event rate (*y*-axis). The dotted diagonal line represents perfect calibration, in which the predicted probabilities align perfectly with the actual event frequencies. The calibration slope for the YOLO-TL model (0.87) was significantly closer to the ideal value of 1 than that of the YOLO-SC model (1.45; *p* = 0.014), indicating a more reliable probability scale. Conversely, the Brier score, which measures the mean-squared prediction error, was slightly more favorable for the YOLO-SC model (0.089 vs. 0.100).

**Figure 6 cancers-17-03144-f006:**
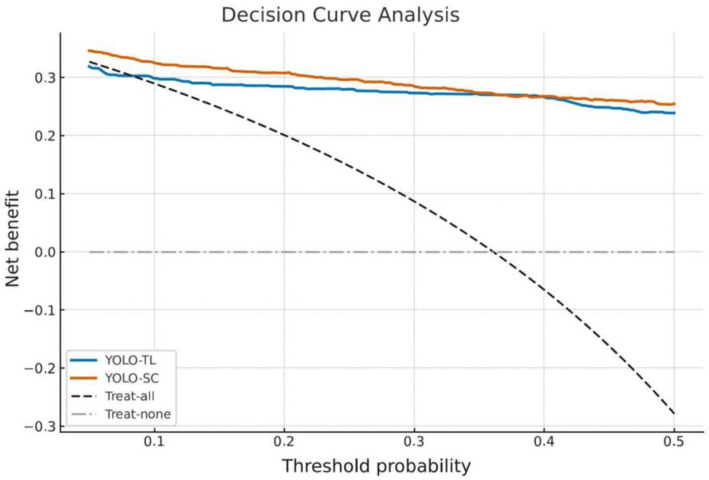
Decision curve analysis comparing the clinical net benefit of the models. The figure shows the decision curve analysis (DCA) for the YOLO-TL (blue line) and YOLO-SC (orange line) models. The *y*-axis represents the net benefit, a metric that weighs the benefit of true-positive decisions against the harm of false-positive decisions, and is plotted across a range of threshold probabilities on the *x*-axis. The performances of the models were compared with two reference strategies: treating all patients (treat-all; black dashed line) and treating no patients (treat-none; gray dash-dotted line). Both models demonstrate a higher net benefit than either reference strategy over a wide range of thresholds, confirming their clinical utility. A direct comparison reveals that the YOLO-SC model provides a marginally, yet statistically significant, higher net benefit than the YOLO-TL model in the approximate threshold probability range of 0.10–0.25.

**Table 1 cancers-17-03144-t001:** Performance comparison at different operating points.

Operating Point	Model	Threshold	Sensitivity	Specificity	PPV	NPV	False Positives	False Negatives
Youden-optimal	YOLO-TL	0.0371	0.914 (0.879–0.944)	0.897 (0.865–0.925)	0.833	0.949	49	23
	YOLO-SC	0.2568	0.892 (0.855–0.925)	0.884 (0.850–0.913)	0.813	0.936	55	29
High Sensitivity (≥0.90)	YOLO-TL	0.0413	0.903 (0.868–0.935)	0.903 (0.873–0.929)	0.840 (0.805–0.872)	0.943 (0.918–0.963)	46	26
	YOLO-SC	0.2230	0.903 (0.868–0.935)	0.867 (0.834–0.896)	0.793 (0.755–0.829)	0.941 (0.916–0.961)	63	26
High Specificity (≥0.90)	YOLO-TL	0.0413	0.798 (0.752–0.840)	0.903 (0.873–0.929)	—	—	46	54
	YOLO-SC	0.2903	0.764 (0.716–0.808)	0.902 (0.872–0.928)	—	—	47	63

## Data Availability

The numeric data presented in Figure 1 are available in Appendix A in the Appendix A. The source code for inference and metric computation is provided as Appendix A in the Appendix A. Raw radiographs and trained model weights are not available.

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
