# Peer review of "Utility of Same-Modality, Cross-Domain Transfer Learning for Malignant Bone Tumor Detection on Radiographs: A Multi-Faceted Performance Comparison with a Scratch-Trained Model"

_cancers, 2025, doi:10.3390/cancers17193144_

Round 1

Reviewer 1 Report

Comments and Suggestions for Authors

Regarding the manuscript titled "Utility of Same-Modality, Cross-Domain Transfer Learning for Malignant Bone Tumor Detection on Radiographs: A Multi-Faceted Performance Comparison with a Scratch-Trained Model", I would like to inform the authors that I reviewed the manuscript and found it to be scientifically valuable. The topic is innovative. The content is well presented. The organization of the content is well done, and it seems that with minimal revisions it can reach excellent quality .

1: The abstract must contain findings. Mention all findings in the abstract.

2: Your manuscript suffers from a major problem, which is the lack of a definition of artificial intelligence and its applications in the field of health. Therefore, I suggest that you provide a brief definition of artificial intelligence and address its numerous applications in health. For this purpose, you can also refer to the following studies that present various applications of machine learning.

"Enhancing Cancer Zone Diagnosis in MRI Images: A Novel SOM Neural Network Approach with Block Processing in the Presence of Noise"
U-Net based estimation of functional connectivity from time series" multi-channel EEG from schizophrenia patients"

Using these resources can mention various applications of machine learning in the health field and also enrich the references of your manuscript.

3: I suggest adding a confusion matrix to this manuscript, which will improve the readability of your study.

4: Be sure to share the implementation code so that we can review it.

The rest of the items are well presented and this study can be proposed for publication after these revisions. So, respond carefully to the revisions.

Author Response

Comments 1: The abstract must contain findings. Mention all findings in the abstract.

Response 1: We thank the reviewer for this helpful suggestion. We have revised the Abstract to explicitly report the principal quantitative findings and their clinical implications while keeping within the 250-word limit. Specifically, we now state: (i) AUCs with 95% CIs and DeLong p-value; (ii) at the high-sensitivity operating point (sensitivity fixed at 0.903 for both models), YOLO-TL’s higher specificity and PPV with corresponding reduction in false positives; (iii) at the high-specificity operating point (~0.90 for both), YOLO-TL’s higher sensitivity; and (iv) YOLO-TL’s superior sensitivity at the Youden-optimal point. To keep the Abstract concise, secondary details (e.g., calibration, DCA) are summarized in the main text and figures.

Comments 2: Your manuscript suffers from a major problem, which is the lack of a definition of artificial intelligence and its applications in the field of health. Therefore, I suggest that you provide a brief definition of artificial intelligence and address its numerous applications in health. For this purpose, you can also refer to the following studies that present various applications of machine learning.

Response 2: Thank you for this valuable suggestion. We agree that the Introduction should briefly define AI/ML and situate our work within the broader landscape of medical imaging. After evaluating the three suggested papers, we found they illustrate AI use in MRI segmentation (SOM) and EEG time-series analysis, but they are not directly aligned with our study’s focus on X-ray imaging and transfer learning. To maintain scientific precision and topical coherence, we revised the Introduction to (i) concisely define AI/ML in radiology and (ii) reference authoritative, peer-reviewed reviews on AI in radiology and transfer learning in medical imaging, as well as recent modality-aligned pretraining evidence (RadiologyNET/foundation models, large-scale TL benchmarks), which collectively underpin our same-modality, cross-domain TL rationale. The new citations include Najjar 2023 and Pinto-Coelho 2023 for AI in radiology, Yu 2022 for TL surveys, Taher 2025 for large-scale TL benchmarking, and Napravnik 2025 for modality-aligned pretraining; we also add modality-specific context for X-ray AI and domain-proximal bone-tumor imaging studies. We believe these sources better support the manuscript’s scope and strengthen its conceptual framing.

Comments 3: I suggest adding a confusion matrix to this manuscript, which will improve the readability of your study.

Response 3: Thank you for the suggestion. Because Table 1 already reports the full confusion-matrix counts (TP, FP, TN, FN) and derived metrics at the high-sensitivity operating point, we added the graphical confusion matrices as Figure S1 to improve readability without duplicating content in the main text. We cite S1 at the end of Section 3.2.3.

Comments 4: Be sure to share the implementation code so that we can review it.

Response 4: Thank you for the suggestion. We now provide the complete inference and evaluation code as Supplementary Software S1–S4 (S1: data loader; S2: YOLO-based model definition; S3: evaluation utilities for per-image scores, ROC/AUC, and confusion matrices; S4: an inference wrapper). To enable full verification of the reported metrics without exposing patient images or proprietary weights, we also release the per-image ground-truth labels and prediction scores used in all analyses (Supplementary Tables S1–S2). In line with our IRB and commercialization constraints, raw radiographs and trained model weights are not available. We believe this package allows reviewers to reproduce all figures and tables derived from prediction scores (e.g., ROC/AUC and operating-point results).

Reviewer 2 Report

Comments and Suggestions for Authors

The manuscript titled “Utility of Same-Modality, Cross-Domain Transfer Learning for Malignant Bone Tumor Detection on Radiographs: A Multi-Faceted Performance Comparison with a Scratch-Trained Model” by Hasei et al. addresses an important challenge in developing medical diagnostic AI systems, that is, the limited availability of high-quality data for large-scale training.

The authors used a well-established approach named “same modality transfer learning” in the novel setting of malignant bone tumor detection on knee radiographs. Even though this method itself is not new, this study’s strength lies in its careful clinical framing, through methodology, and comprehensive evaluation. Taken together, this study provides valuable evidence for the utility of the same modality transfer learning in rare disease diagnosis, where access to high-quality data is particularly limited.

The authors compared the YOLO-based model pretraining on the chest X-rays (YOLO-TL) with a model trained from scratch (YOLO-SC) on bone radiographs using internal and external datasets for an unbiased assessment. Even though overall AUC performance is similar, YOLO-TL demonstrated superior specificity and positive predictive value at clinically critical high-sensitivity thresholds.

The study design and methodology are strong, with a separate internal validation set and an external validation data set, and robust evaluation metrics (ROC, PR curves, calibration, decision curve analysis)

The manuscript is well-written with clear, easy-to-understand plots.

It would be beneficial if authors further discuss results, such as the small net-benefit advantage of YOLO-SC in certain ranges, particularly in terms of practical implications.

Overall, this study makes a meaningful contribution to the field of early diagnosis of malignant bone tumors.

Author Response

Comment: It would be beneficial if authors further discuss results, such as the small net-benefit advantage of YOLO-SC in certain ranges, particularly in terms of practical implications.

Response: We appreciate this helpful suggestion. We have added a brief clarification to the Discussion noting that YOLO-SC shows a small net-benefit advantage only within a limited probability-threshold range (≈0.10–0.25) (see Fig. 6), which may be relevant for workflows that explicitly adopt a threshold-based triage rule. However, our pre-specified clinical use case prioritizes a high-sensitivity operating point to minimize missed malignant tumors; in that context, YOLO-TL reduces false positives at identical sensitivity. We therefore interpret the DCA difference as modest and context-dependent, and it does not change our main conclusion.